# Chronic peptide-based GIP receptor inhibition exhibits modest glucose metabolic changes in mice when administered either alone or combined with GLP-1 agonism

**Jason A. West**[1], **Anastasia Tsakmaki**[2], **Soumitra S. Ghosh**[3], **David G. Parkes**[4], **Rikke V. Grønlund**[5], **Philip J. Pedersen**[5], **David Maggs**[1¤], **Harith Rajagopalan**[1], **Gavin A. Bewick**[2]*

**1** Fractyl Laboratories Inc, Lexington, MA, United States of America, **2** Diabetes Research Group, School of Life Course Sciences, Faculty of Life Science and Medicine, King's College London, London, England, United Kingdom, **3** Doon Associates LLC, San Diego, CA, United States of America, **4** DGP Scientific Inc, San Diego, CA, United States of America, **5** Gubra ApS, Hørsholm, Denmark

¤ Current address: Becton Dickinson Technologies & Innovation, Raleigh-Durham, NC, United States of America

* gavin.bewick@kcl.ac.uk

**Data Availability Statement:** All relevant data are within the manuscript and its Supporting Information files.

## Abstract

Combinatorial gut hormone therapy is one of the more promising strategies for identifying improved treatments for metabolic disease. Many approaches combine the established benefits of glucagon-like peptide-1 (GLP-1) agonism with one or more additional molecules with the aim of improving metabolic outcomes. Recent attention has been drawn to the glucose-dependent insulinotropic polypeptide (GIP) system due to compelling pre-clinical evidence describing the metabolic benefits of antagonising the GIP receptor (GIPR). We rationalised that benefit might be accrued from combining GIPR antagonism with GLP-1 agonism. Two GIPR peptide antagonists, GIPA-1 (mouse GIP(3–30)NH2) and GIPA-2 (NαAc-K10[γEγE-C16]-Arg18-hGIP(5–42)), were pharmacologically characterised and both exhibited potent antagonist properties. Acute *in vivo* administration of GIPA-1 during an oral glucose tolerance test (OGTT) had negligible effects on glucose tolerance and insulin in lean mice. In contrast, GIPA-2 impaired glucose tolerance and attenuated circulating insulin levels. A mouse model of diet-induced obesity (DIO) was used to investigate the potential metabolic benefits of chronic dosing of each antagonist, alone or in combination with liraglutide. Chronic administration studies showed expected effects of liraglutide, lowering food intake, body weight, fasting blood glucose and plasma insulin concentrations while improving glucose sensitivity, whereas delivery of either GIPR antagonist alone had negligible effects on these parameters. Interestingly, chronic dual therapy augmented insulin sensitizing effects and lowered plasma triglycerides and free-fatty acids, with more notable effects observed with GIPA-1 compared to GIPA-2. Thus, the co-administration of both a GIPR antagonist with a GLP1 agonist uncovers interesting beneficial effects on measures of insulin sensitivity, circulating lipids and certain adipose stores that seem influenced by the degree or nature of GIP receptor antagonism.

**Funding:** This study was supported by Fractyl Laboratories Inc., Lexington, MA. Employees and shareholders of Fractyl Laboratories Inc played a prominent role in the study design, data collection and analysis, decision to publish, and preparation of the manuscript. All authors reviewed and critiqued the manuscript throughout the editorial process, approved the final manuscript draft submitted for publication, and agreed to be accountable for all aspects of the work, ensuring the accuracy and integrity of the publication.

**Competing interests:** The authors have read the journal's policy and have the following competing interests: J.A.W. and H.R. are employees and shareholders of Fractyl Laboratories Inc. A.T. has received funding/grant support from the Juvenile Diabetes Research Foundation (JDRF). S.S.G. is an employee of Doon Associates and has received honorariums for consultancy from Fractyl Laboratories Inc. D.G.P. is an employee of DPB Scientific and has received honorariums for consultancy from Fractyl Laboratories Inc R.V.G. and P.J.P. are employees of Gubra ApS. D.M. is an ex-employee of Fractyl Laboratories Inc., is a current shareholder, and has received honorarium for consultancy from Fractyl Laboratories Inc. G.A. B. has received funding/grant support from the European Foundation for the Study of Diabetes and JDRF and honorarium for consultancy from Fractyl Laboratories Inc. This does not alter our adherence to PLOS ONE policies on the sharing of data and materials. There are no patents, products in development or marketed products associated with this research to declare.

## Introduction

The global obesity and diabetes crises have prompted a concerted effort to identify effective novel treatments. Gut hormones exhibit well-characterised physiological roles ranging from effects on pancreatic islet hormone secretion, glucose concentrations, lipid metabolism, energy storage, gut motility, appetite, and body weight. These properties have drawn attention to their potential to treat metabolic disease, and the glucagon-like peptide-1 (GLP-1) drug class is now a well-established treatment for Type 2 diabetes (T2D) and obesity. Current investigational therapeutic strategies centre on designing drugs based on gut hormones that provide synergistic or additive effects in combination but questions remain regarding which combination of hormones will produce the most effective, safe and tolerated treatment for obesity and diabetes.

Two key glucoregulatory gut hormones, GIP and GLP-1, are secreted respectively by K cells, located predominantly in the proximal small intestine, and by L cells, most densely located in the distal small intestine and colon [1]. Both hormones are secreted following nutrient intake and augment insulin secretion. Together they produce the incretin effect, a 2–3 fold increase in insulin production in response to oral glucose compared with the equivalent glucose dose administered intravenously [2, 3], suggesting a combination of these two hormones could be beneficial for the treatment of T2D.

Indeed, GIP is the major incretin hormone in healthy individuals, responsible for around 70% of the effect [4]. However, individuals with T2D exhibit an impaired GIP insulinotropic effect [5] and increased fasting plasma GIP concentrations [6]. In contrast, administration of exogenous GLP-1 results in normalization of fasting hyperglycaemia in T2D patients [7]. Moreover, GLP-1 is a potent inhibitor of appetite, food intake [8] and glucagon secretion [9]. As a result, GLP-1 monotherapy is a key therapeutic modality for T2D therapy [10].

Despite the introduction of GLP-1 agonists into the T2D treatment paradigm, well-controlled diabetes management remains elusive and many patients progress to insulin therapies. Weight-loss surgeries like Roux-en-Y gastric bypass (RYGB) and vertical sleeve gastrectomy are the only treatments associated with T2D remission [11–15]. Among the proposed mechanisms contributing to the metabolic benefits of RYGB are alterations in gut hormone secretion and sensitivity [16]. For example, post-prandial plasma GLP-1 levels dramatically increase after RYGB surgery [17], as GLP-1 secretion is increased in the distal regions of the small intestine due to increased nutrient delivery [18]. However, contradictory reports regarding GIP secretion following RYGB exist; some report a reduction in GIP [19, 20], others no alteration [21] whilst others an increase in secretion [22].

Growing evidence suggests that GIP may act as an obesogenic factor under certain circumstances, further complicating rational design of combination therapies [23]. The GIP receptor (GIPR) is expressed in the pancreas but is also found in adipocytes [24] and brain [25], which are important regulators of body weight. GIP increases the activity of lipoprotein lipase [26], elevating triglyceride accumulation in fat tissue [27] and inhibiting lipolysis [28]. Moreover, in healthy individuals, GIPR has been identified in multiple genome wide association studies as a contributing factor to obesity in European populations [29, 30]. Obesogenic diets cause hyperplasia of K cells and increase their density, driving elevated circulating GIP concentrations [31]. Finally, GIP, K cell, and GIPR knockout mice are resistant to high-fat diet (HFD)-induced obesity and insulin resistance [32–34]. All the above evidence warrants further investigation and has led to discovery programs focused on blocking GIP activity [23, 35]. Several different GIPR antagonists have been identified including vaccines against GIP [36, 37], GIP neutralising antibodies [38], antibodies against the GIPR [39, 40] and antagonist peptides [41–44] (for a detailed summary of these studies see [35]). However, pharmacological approaches have not yielded satisfactory data and have been controversial.

A monoclonal GIP neutralising antibody prevented weight gain independently of food intake in a mouse model of diet-induced obesity (DIO), reduced fasting insulin concentrations and improved intraperitoneal glucose tolerance [38]. Peripheral administration of gipg013 (another GIPR neutralizing antibody) to obese mice prevented body weight gain, and when administrated centrally, reduced body weight and adiposity, likely through a leptin sensitizing mechanism [45]. In a second study, gipg013 reduced food intake and weight gain in high-fat fed mice, and improved glucose tolerance [46]. However, weight loss was not enhanced following coadministration of gipg013 with a GLP-1 agonist. Surprisingly, the combined antagonism of GIPR and GLP-1R produced additional protection against DIO. In one of the most promising studies, Killion *et al.* demonstrated that peripheral administration of murine anti-GIPR antibody prevented DIO in mice and was associated with reduced food intake, improved metabolism and decreased resting respiratory exchange ratio [40]. Excitingly, similar findings were observed in obese nonhuman primates (NHPs) utilizing an anti-human GIPR antibody. Additionally, enhanced weight loss was observed in both mice and NHPs when anti-GIPR antibodies were co-administered with GLP-1R agonists. However, GIPR antagonism did not result in improved glucose tolerance.

Further controversy is evident with peptide based GIPR antagonists. GIP(3–30)NH$_2$, an N-terminally truncated form of GIP has been described as a potent peptide-based antagonist [42, 47, 48]. Sub-chronic dosing of rat GIP(3–30)NH$_2$ increased body weight, fat mass, triglycerides, LPL, and leptin levels in rats fed normal chow [49]. A fatty-acylated, N-terminally truncated GIP analogue with high *in vitro* antagonism potency for mouse GIPR had no effect on body weight whether dosed alone or in combination with liraglutide in DIO mice [50]. In contrast, a palmitoyl variant of Pro$^3$GIP(3–30) with a C-terminal extension enhanced insulin sensitivity and reduced body weight as well as circulating glucose and insulin in DIO mice [41].

Previous studies have demonstrated that GIPR blockade increases sensitivity to GLP-1R agonism at least at the level of pancreatic beta cells [51, 52]. Killion *et al.* revealed chronic antagonism of GIPR enhanced GLP-1R agonist-driven weight loss [40], even though similar effects were not reproduced with gipg013 antibody [46]. In addition to these studies, there is an additional body of literature supporting GIPR agonism and not antagonism as a therapeutic strategy for T2D [23, 53, 54], and clinical testing is ongoing for a dual GIPR/GLP-1R agonist peptide [55, 56]. Given the conflicting results surrounding the co-targeting of GIP and GLP-1 signalling pathways, we investigated the metabolic benefits of co-administration of peptide-based GIPR antagonists with a GLP-1 agonist, liraglutide. We used two GIPR antagonists; mouse GIP(3–30)NH$_2$ (GIPA-1) as well as N$^\alpha$Ac-K10[γEγE-C16]-Arg18-hGIP(5–42) (GIPA-2) [50]. *In vitro* functional activities of the peptides were confirmed before the metabolic consequences of acute and chronic dosing in mice with or without the GLP-1R agonist liraglutide were determined. While GLP-1 agonism via liraglutide demonstrated the expected pharmacodynamic actions with chronic treatment of DIO mice, including weight loss, reduced food intake and reduced fasting blood glucose as well as improved insulin sensitivity, the validated GIPR peptide antagonists either alone or in combination showed modest effects on insulin sensitivity and adiposity in this study, underscoring the complex and multifaceted control of glucose homeostasis and metabolism *in vivo*.

## Materials and methods

### Peptides

Two GIPR antagonist peptides were manufactured by CPC Scientific Inc.:

GIPA-1 (mouse GIP(3–30)NH$_2$) was selected for investigation in mouse studies as it is species specific. It has the sequence: Glu-Gly-Thr-Phe-Ile-Ser-Asp-Tyr-Ser-Ile-Ala-Met-Asp-Lys-Ile-Arg-Gln-Gln-Asp-Phe-Val-Asn-Trp-Leu-Leu-Ala-Gln-Arg-NH$_2$.

GIPA-2 (N$^{\alpha}$Ac-K10[γEγE-C16]-Arg18-hGIP(5–42)) has been reported as a potent antagonist for mouse GIPR, with no cross-reactivity at mouse-derived GLP-1 or glucagon receptors [50]. It has the sequence: Ac-Thr-Phe-Ile-Ser-Asp-Lys(isoGlu-isoGlu-palm)-Ser-Ile-Ala-Met-Asp-Lys-Ile-Arg-Gln-Gln-Asp-Phe-Val-Asn-Trp-Leu-Leu-Ala-Gln-Lys-Gly-Lys-Lys-Asn-Asp-Trp-Lys-His-Asn-Ile-Thr-Gln.

Liraglutide was used for GLP-1 receptor agonism (Novo Nordisk).

Mouse GIP was sourced from Phoenix Pharmaceuticals.

### *In vitro* receptor activity/ cAMP measurements

Each peptide was tested for its ability to activate or inhibit the mouse GIP receptor using the cAMP Hunter™ CHO-K1 GIPR Gs cell line, in combination with the Hit Hunter® cAMP XP + assay for determining cAMP, according to the manufacturer's instructions (Eurofins DiscoverX Corporation).

### Animals

All animal experiments were performed according to the bioethical guidelines of Gubra (Hørsholm, Denmark) under the personal license 2017-15-0201-01378, which was issued by the Danish Animal Experimentation Council. The Gubra IACUC approved all animal study designs including the associated animal care conducted in this report and upheld all regulatory compliance. Male C57Bl/6JRj mice between 5–7 weeks old were purchased from Janvier Labs. All mice were acclimated for at least one week before the initiation of all studies. Mice were kept at 22 ± 2°C on a 12:12-h light-dark cycle and had access to food (Altromin #1324 [11% fat, 24% protein, 65% carbohydrate], Brogaarden) and water *ad libitum* unless noted otherwise. The animal monitoring consisted of daily visual inspections and routine weight monitoring, as well as food intake measurements during the chronic dosing studies. If animals were found in a critical condition in the study period (serious or severe side effect), euthanization of the affected animal occurred without further delay. The following humane endpoints were used for the current studies: Rapid weight loss of ≥20 percent within a few days (<5), Gradual weight loss over a longer period leading to emaciation (the limit is 20 percent below the weight of a normal healthy control animal of the same species and age), clinical or behavioural signs such as inactivity and loss of interest in the surroundings, inability to access food or water, forced abdominal respiration, Dehydration leading to reduced skin elasticity, the presence of deep open wounds or large tumours, Swelling, redness and/or pain response from tissue on and around the osmotic pump, dehiscence (splitting apart) of surgical incisions from pump implantations which cannot be re-sutured or discharge from surgical incisions, any condition or test compound indicated to cause suffering in the animals, Local reactions to compound injections causing inflammation or wounds of the cutis, and/or any other adverse reactions to the compound resulting in any of the above-mentioned signs of pain and distress. Animals included in the acute *in vivo* studies were euthanized by cervical dislocation. Animals included in the GIPA-1 and GIPA-2 studies were euthanized be cardiac bleeding under isoflurane/O$_2$ inhalation anaesthesia.

### Acute *in vivo* studies

The acute effects of GIPA-1 and GIPA-2 on glucose homeostasis and insulin secretion were studied in 10-week old lean male C57BL/6JRj mice. Animals (n = 8 for each group) fasted for 4 hours, were dosed via subcutaneous injection with 1, 5 or 10 mg/kg of either GIPA-1 15

minutes before oral glucose administration (1.5 g/kg), or GIPA-2 60 minutes before oral glucose administration (1.5 g/kg). Blood glucose was measured just before the administration of GIPA-1 or GIPA-2 and at 0, 15, 30 60, 120, 180 and 240 min. Plasma insulin was measured at 15 min.

The effects of GIPA-2 and mouse GIP alone or in combination were studied in 10-week old lean male C57BL/6JRj mice. Animals were fasted 4 hours prior to an intraperitoneal glucose bolus (1.5 g/kg). Sixty minutes prior to glucose load animals received subcutaneous injection of 5mg/kg GIPA-2, while animals receiving mouse GIP (0.1 mg/kg) were dosed subcutaneously 15 minutes prior to glucose load. Blood glucose was measured just before the administration of GIPA-2 and at 0, 15, 30 60, 120, 180 and 240 min. Plasma insulin was measured at 0, 15 and 240 min.

## Chronic *in vivo* studies

**GIPA-1 study.** 23-week-old male C57BL/6JRj mice (n = 40), fed a HFD (SSNIFF® diet #12492[60% fat, 20% protein, 20% carbohydrate]) for 14 weeks, were distributed into 4 test groups and received daily either vehicle, liraglutide (0.2 mg/kg/day), GIPA-1 (~4.5 mg/kg/day) or liraglutide + GIPA-1 for 4 weeks while still maintained on HFD. GIPA-1 was delivered using minipumps in order to achieve sufficient GIPR antagonism based on the properties of this peptide (Fig 2A). Liraglutide was dissolved in phosphate-buffered saline (PBS) plus 0.1% bovine serum albumin (BSA) (vehicle 1) and GIPA-1 in DMSO/propylene glycol (50/50 volume per volume) (vehicle 2). 100 μL minipumps (Alzet) were filled with GIPA-1 (30 mg/mL) or vehicle 2 and kept in 0.9% saline at 37˚C overnight. The minipump was placed subcutaneously through a 1-cm long incision in the neck with the animal under isoflurane anesthesia. Animals were treated with analgesics on day 0–3 (NSAID, carprofen 50 mg/kg SC QD). From day 0 (first dose) through day 28, mice received one of the following four treatments per day: (1) vehicle 1 (subcutaneous injection (SC)) and continuous infusion of vehicle 2 via osmotic minipump; (2) 0.2 mg/kg liraglutide (SC) and continuous infusion of vehicle 2 via osmotic minipump; (3) vehicle 1 (SC) and continuous infusion of GIPA-1 (~4.5 mg/kg/day; 56.8 nmol/kg/h) via osmotic minipump; and (4) liraglutide (0.2 mg/kg liraglutide, SC) plus continuous infusion of GIPA-1 (~4.5 mg/kg/day) via osmotic minipump. Dose levels of GIPA-1 were based on targeting plasma exposure levels sufficient to antagonise endogenous GIP and selected based on *in vitro* antagonism data. Fasted (4 hrs) blood glucose and insulin levels were measured on study day -3, 14 and 27. OGTT was performed on day 21 of the study. On the day of the OGTT the fasted animals received their daily doses of vehicles and liraglutide thirty minutes before receiving a glucose bolus (2 g/kg). Blood glucose was measured before their daily dosing and at 0, 15, 30 60, 120, 180 min after glucose administration. Plasma insulin was measured at 0 and 15 min. Whole body weight and food intake were measured throughout the study period. On the termination day (day 28), liver, epididymal fat depot, mesenteric fat depot, retroperitoneal fat depot and inguinal fat depot were dissected and weighted. Blood collected from the tail vein was used for determination of free fatty acids (FFAs) plasma concentration and blood collected via cardiac puncture was used for determination of terminal plasma triglyceride (TG) and total cholesterol (TC).

**GIPA-2 study.** For studying the chronic effects of GIPA-2, male C57BL/6JRj mice (n = 40) were fed a HFD (SSNIFF® diet #12492) for 43 weeks. DIO mice were randomized (n = 10 per group) based on baseline blood glucose and body weight. Liraglutide was dissolved in PBS plus 0.1% BSA (Vehicle1) and GIPA-2 in PBS (vehicle 2). Mice received one of the following four treatments daily for 28 days while still maintained on HFD: (1) vehicle 1 (SC) and vehicle 2 (SC); (2) 0.2 mg/kg liraglutide (SC) and vehicle 2; (3) vehicle 1 (SC) and 10 mg/kg

GIPA-2 (SC); and (4) 0.2 mg/kg liraglutide (SC) plus 10 mg/kg GIPA-2 (SC). GIPA-2 dose levels were selected based on doses shown to sufficiently antagonize insulinotropic actions of exogenous GIP in acute IPGTT studies (Fig 1J) adjusted for single daily dosing over the 4-week treatment period. Fasted (4 hrs) blood glucose and insulin were measured on study day -4, 10 and 26. An OGTT was performed on day 17. On the day of the OGTT the fasted animals (4 hrs) received their daily doses of vehicles, liraglutide and GIPA-2 one hour before receiving a glucose bolus (2 g/kg). Blood glucose was measured before their daily dosing and at 0, 15, 30 60, 120, 180 and 240 min after glucose administration. An insulin tolerance test (ITT) was performed on day 24. Fasting (4 hrs) mice received their daily doses of vehicles, liraglutide and GIPA-2 one hour before receiving insulin (0.5 U/kg, intraperitoneal injection). Blood glucose was measured before their daily dosing and at 0, 15, 30 60, 120, 180 min after glucose administration. Similar measurements were performed as in the GIPA-1 study. One animal from the vehicle group was found dead on study day 28. At necropsy the animal was found have a liver tumor, which as attributed to both the age of the animal (52 weeks) and the DIO diet.

## Blood pharmacology

Fasting blood glucose was measured using a BIOSEN c-Line glucose meter (EKF-Diagnostics), insulin was measured using the Meso Scale Diagnostics platform, TG and TC were measured using commercial kits (Roche Diagnostics) on the Cobas C-501 autoanalyzer (Roche), and FFA levels were measured using a Wako Chemicals kit on the Cobas C-501 autoanalyzer, all according to the manufacturers' instructions. Homeostatic model assessment of insulin resistance (HOMA-IR) was calculated using the following equation: [fasting serum glucose (mmol/L) × fasting serum insulin (pmol/L) / 22.5] to assess insulin resistance [57].

## Statistical analysis

All statistical analyses were performed using Graph Pad Prism Version 8.1.2 (GraphPad Software). Results are expressed as mean ± standard error of the mean (SEM). Statistical significance was evaluated using one-way or two-way analysis of variance (ANOVA), as appropriate, followed by Tukey's or Dunnett's post hoc test. Relevant tests are described in figure legends. A cut off of $p < 0.05$ was used as a threshold for statistical significance.

## Results

### *In vitro* receptor activity of GIPR peptide antagonists

We began by characterising the receptor-interaction properties of GIPA-1 and GIPA-2. Their agonist and antagonist potencies were measured via a cAMP production assay in CHO-K1 cells overexpressing the mouse GIPR. Neither GIPA-1 or GIPA-2 exhibited agonist properties against the mouse GIPR (S1A and S1B Fig). In antagonist assays, GIPA-2 was more potent in comparison to GIPA-1 at blocking mouse GIPR activation by mouse GIP for cAMP production with $IC_{50}$ values of 3 nM and 483 nM respectively (Fig 1A and 1B).

### Acute *in vivo* effects of GIPR peptide antagonists in lean mice

Having established both GIPA-1 and GIPA-2 as potent mouse GIPR antagonists *in vitro*, we aimed to determine their acute actions on oral glucose tolerance in lean mice.

Acute treatment with 1, 5 or 10 mg/kg of GIPA-1 had no effect on glucose tolerance (Fig 1C and 1D) or plasma insulin concentrations 15 minutes post oral glucose bolus (Fig 1G). In contrast, delivery of the more potent GIPA-2 produced a dose dependent increase in glucose

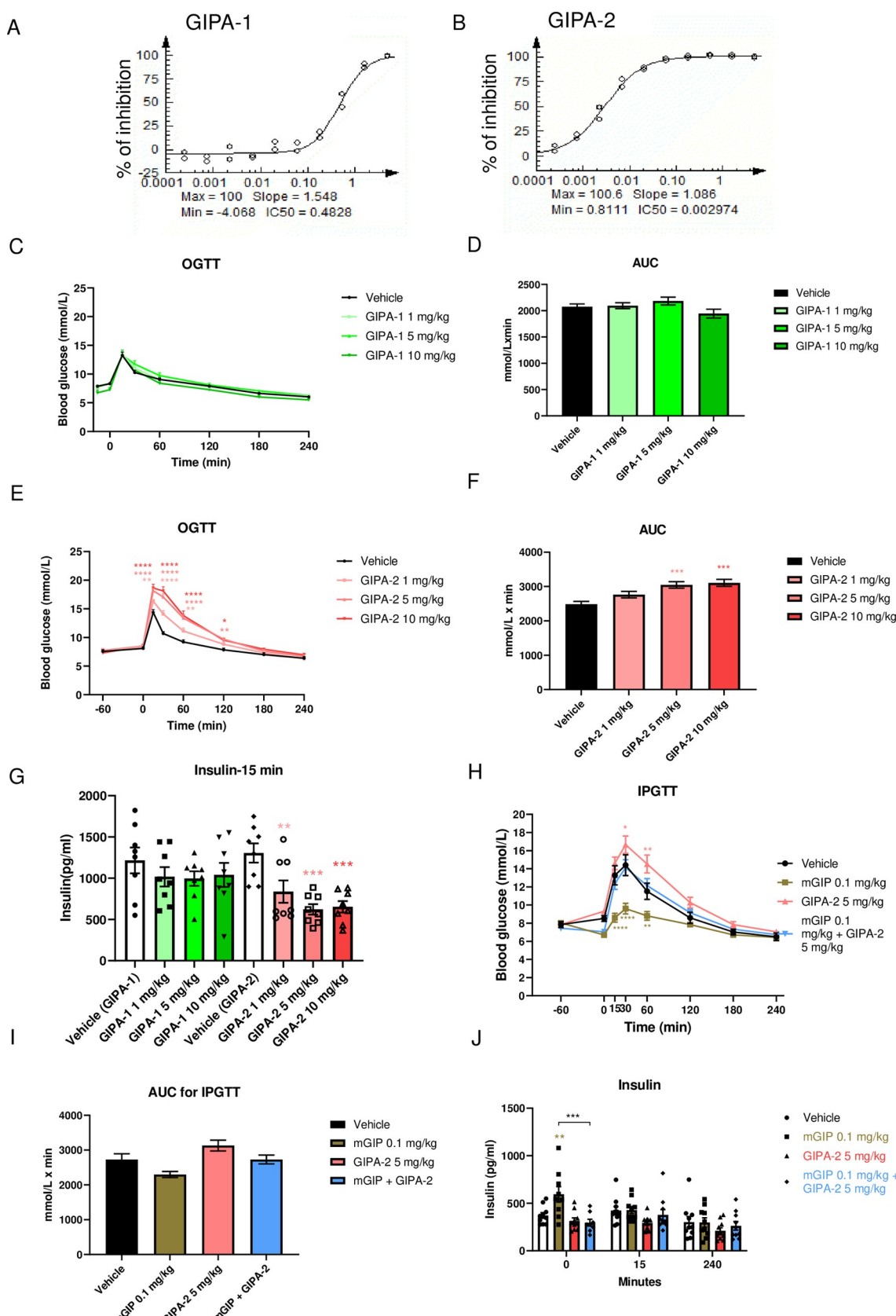

**Fig 1. Effects of acute GIPR antagonism on glucose homeostasis and insulin secretion.** GIPA-1 and GIPA-2 are antagonists of mouse GIPR. Plots (A and B) shows the inhibition of mouse GIPR by GIPA-1 (A) and GIPA-2(B) in CHO-K1 cells. $IC_{50}$ values (μM) are indicated on the plots. (C-J) Acute effects of GIPA-1 and GIPA-2 on glucose homeostasis and insulin secretion in lean mice. Oral glucose tolerance tests (C and E), the corresponding areas under the curve (AUC) (D and F) and insulin responses (I) following subcutaneous injections of vehicle, 1,5 and 10 mg/kg GIPA-1 (C and D) or vehicle, 1,5 and 10 mg/kg GIPA-2 (E and F) before an oral glucose administration (1.5 g/kg). (G, H and J) Acute effect of GIPA-2 alone or in combination with mouse GIP on glucose homeostasis and insulin secretion in lean mice. Intraperitoneal glucose tolerance test cute IPGTT (G), the corresponding AUC (H) and insulin response (J) following subcutaneous injections of vehicle, 0.1 mg/kg mouse GIP, 5mg/kg GIPA-2, and combination of both. Group sizes are n = 8, and data are represented as mean ± SEM. Statistical analysis was calculated using two-way ANOVA with Tukey's post-hoc test (C, E, G and J) and one-way ANOVA with Dunnett's post-hoc test (D, F, H, I). *: p< 0.05, **: p < 0.01, ***: p < 0.001, ****: p<0.0001 compared to vehicle.

area under the curve (AUC) (Fig 1E and 1F). This was associated with an inhibition of glucose stimulated insulin secretion as measured by plasma insulin concentrations 15 min post glucose bolus (Fig 1G).

To determine if GIPA-2's effects on glucose tolerance were solely driven by its ability to block mouse GIPR signalling, we co-administered GIPA-2 with mouse GIP during an intraperitoneal glucose tolerance test (IPGTT). As expected, acute treatment with mouse GIP significantly improved glucose tolerance, whilst GIPA-2 significantly decreased glucose tolerance (Fig 1H and 1I). A robust increase in plasma insulin was observed 15 minutes after mouse GIP administration, just prior to glucose bolus (0 min timepoint). This effect was not observed at the 15 and 240 min timepoints (Fig 1J). Importantly, combined treatment with GIPA-2 and mouse GIP resulted in attenuation of glucose tolerance (Fig 1H and 1I) compared to mouse GIP treatment and an inhibition of mouse GIP stimulation of insulin secretion (Fig 1J), demonstrating GIPA-2 to be an effective antagonist of exogenous GIP actions *in vivo*.

## Chronic *in vivo* effects of GIPR peptide antagonists alone and in combination with GLP-1R agonist in DIO mice

We used a mouse model of DIO and insulin resistance to investigate the potential chronic metabolic benefits of GIPA-1 and GIPA-2 administration, alone or in combination with the GLP-1 receptor agonist liraglutide. DIO mice were treated for 28 days and food intake, body weight, and fasting glucose and insulin concentrations were measured at regular intervals (Fig 2A).

Chronic administration of GIPA-1 or GIPA-2 alone had no effect on absolute body weight (Fig 2B and 2C) or cumulative food intake (Fig 2D and 2E) compared with vehicle control. As expected, liraglutide reduced both body weight and food intake, but no additive effect was observed by combining liraglutide with either of the antagonists on these parameters.

Fasting blood glucose and plasma insulin concentrations were similar between groups at the beginning of the studies (Fig 2F–2I). In agreement with our acute dosing data, chronic dosing of GIPA-1 alone did not alter fasting blood glucose at any time point (Fig 2F), whilst liraglutide produced significant reductions for the duration of the study. In combination, liraglutide plus GIPA-1 also reduced fasting blood glucose but the combination was only significantly better at reducing blood glucose than liraglutide alone on day 14. Associated plasma insulin was reduced in both the liraglutide and combination groups. This was significant for the combination, but not for liraglutide monotherapy (Fig 2H).

GIPA-2 modestly reduced fasting glucose concentrations only being significant on day 10 (Fig 2G). In combination with liraglutide, GIPA-2 had no additive effect on the ability of liraglutide to lower fasting glucose concentration (Fig 2G). Fasting insulin concentrations were not significantly different between groups (Fig 2I).

Interestingly, the one area where the combinatorial effects of pharmacologic inhibition of GIPR and GLP-1R agonism showed some separation from the single intervention arm (GLP-1

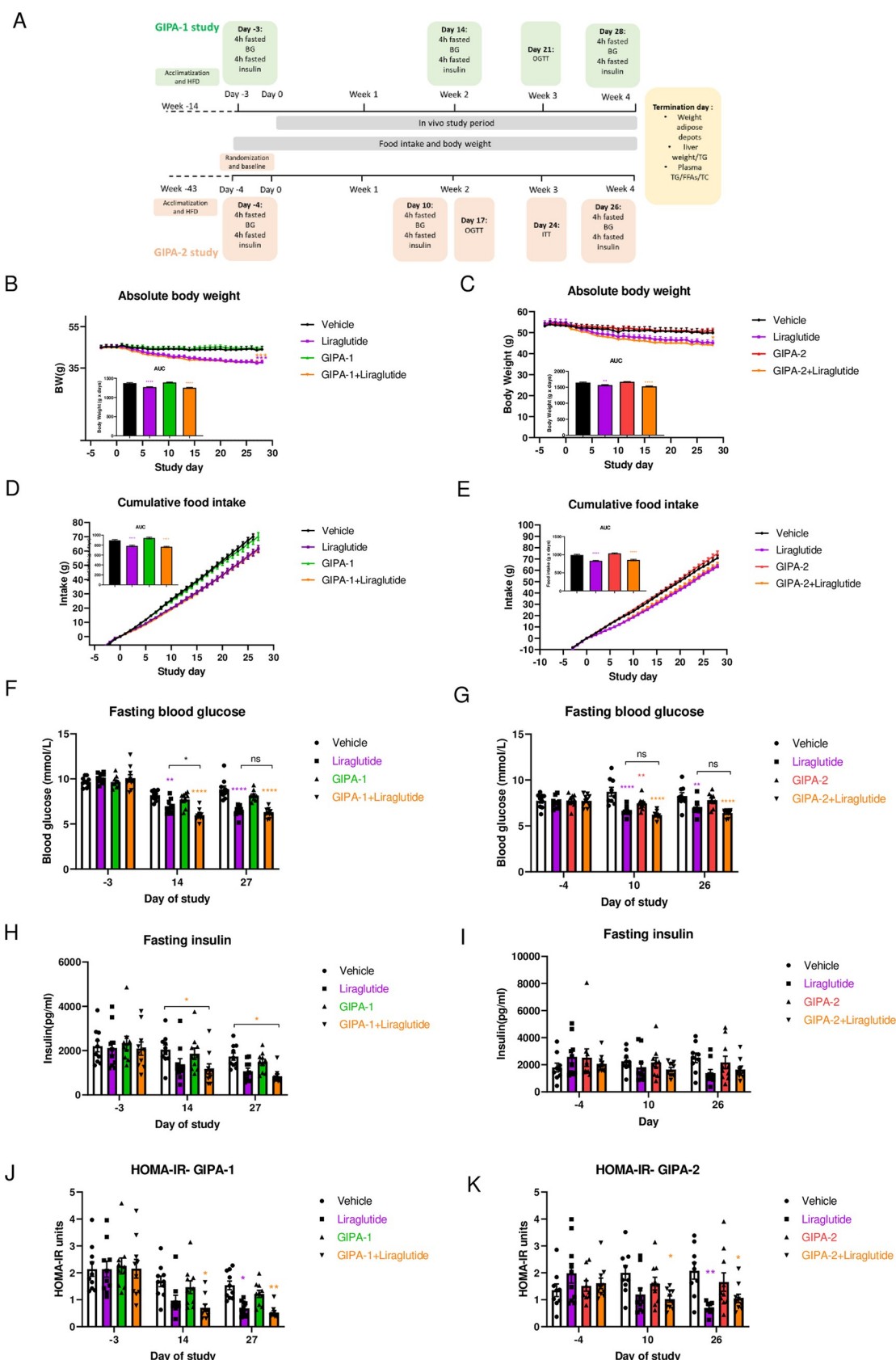

**Fig 2. Effects of chronic GIPR antagonism on body weight, food intake, and fasting blood glucose and insulin.** (A) GIPA-1 and GIPA-2 chronic studies schematic. (B-K) Effects of chronic administration of GIPA-1 and GIPA-2 on body weight (B and C), food intake (D and E), fasting plasma glucose (F and G), fasting plasma insulin (H and I) and HOMA-IR (J and K) in DIO mice. Group sizes are n = 10, and data are represented as mean ± SEM. Statistical analysis was calculated using two-way ANOVA with Tukey's post-hoc test and one-way ANOVA for AUC with Dunnett's post-hoc test. *: p< 0.05, **: p < 0.01, ***: p < 0.001, ****: p<0.0001 compared to vehicle.

alone) was with regard to plasma insulin levels and calculations of homeostatic model assessment of insulin resistance (HOMA-IR). On days 14 and 27 of the GIPA-1 study, GIPA-1 combined with liraglutide reduced HFD-induced hyperinsulinemia and insulin resistance (Fig 2J). In GIPA-2 study, on day 10, only the combination therapy showed a reduction in insulin resistance, whereas on day 26 both liraglutide monotherapy and combination therapy demonstrated a similar reduction in HOMA-IR compared to vehicle control (Fig 2K).

The effect of GIPR antagonism on oral glucose tolerance was tested mid-study. Whereas GIPA-1 and GIPA-2 monotherapy resulted in elevated glucose excursions, their administration with liraglutide did not adversely impact the beneficial glucose lowering effects of liraglutide (Fig 3A, 3B, 3E and 3F). In the GIPA-1 study, baseline blood glucose (-30 min) (Fig 3A) was lower in the liraglutide group and in the combination group, in line with our previous results (Fig 2F), but this effect was not evident for GIPA-2 (Fig 3E).

Fasting insulin concentrations tracked the glucose trend for GIPA-1, with no effect compared to vehicle control when given alone, and no additional reduction of liraglutide-induced decrement in insulin at time 0 of the OGTT (Fig 3C). A similar non-significant pattern of results was apparent at 15 minutes post glucose administration (Fig 3D).

In the GIPA-2 study, we assessed insulin sensitivity using an ITT protocol on day 24. GIPA-2 monotherapy had no effect on insulin sensitivity and when delivered in combination with liraglutide appeared to negatively impact the improved insulin sensitivity produced by liraglutide (Fig 3G and 3H).

Finally, we studied the chronic effects of GIPR antagonism on circulating lipids (total cholesterol (TC), triglycerides (TG) and free fatty acids (FFA)) and white adipose depots (including epididymal, mesenteric, inguinal, and retroperitoneal fat). We also accounted for fat deposition in key internal organs through measures of pancreas, kidney, duodenum, jejunum-ileum and liver weight. Notably, the combination of GIPR antagonism via GIPA-1 and GLP-1 agonism had effects to lower plasma TG (Fig 4A) plasma FFA (Fig 4C), and to cause a reduction in epididymal fat mass (Fig 4G) GIPA-2 showed similar reduction in epididymal fat mass on combination with liraglutide (Fig 4H) when compared with single intervention arms. While this is consistent with reports of GIP's role in the regulation of adiposity, we did not observe similar and consistent effects of GIPR antagonism on circulating lipids and other fat depots. Monotherapy with either of the GIPR antagonists or liraglutide had no effect on either plasma TG or concentrations of circulating FFA (Fig 4A–4D). Liraglutide alone significantly reduced circulating TC and liver TG concentrations but GIPR antagonism did not (Figs 4E, 4F, S2A and S2B). There was no impact on liver weight in any group (S2C and S2D Fig). While the combination elicited a significant impact on TC concentrations (Fig 4E and 4F) and liver TG (S2A and S2B Fig), this effect was not superior to that of liraglutide monotherapy (Fig 4E). Mesenteric fat weight was reduced by combination therapy in the GIPA-1 study, but no additive effect was apparent (Fig 4I). In the GIPA-2 study, mesenteric fat weight was unaffected by any treatment (Fig 4J). Retroperitoneal fat weight was reduced in the GIPA-1 study in both the liraglutide and combination groups but was again unaffected in the GIPA-2 study (S2E and S2F Fig). There were no differences on inguinal fat weight between groups in either study (S2G and S2H Fig).

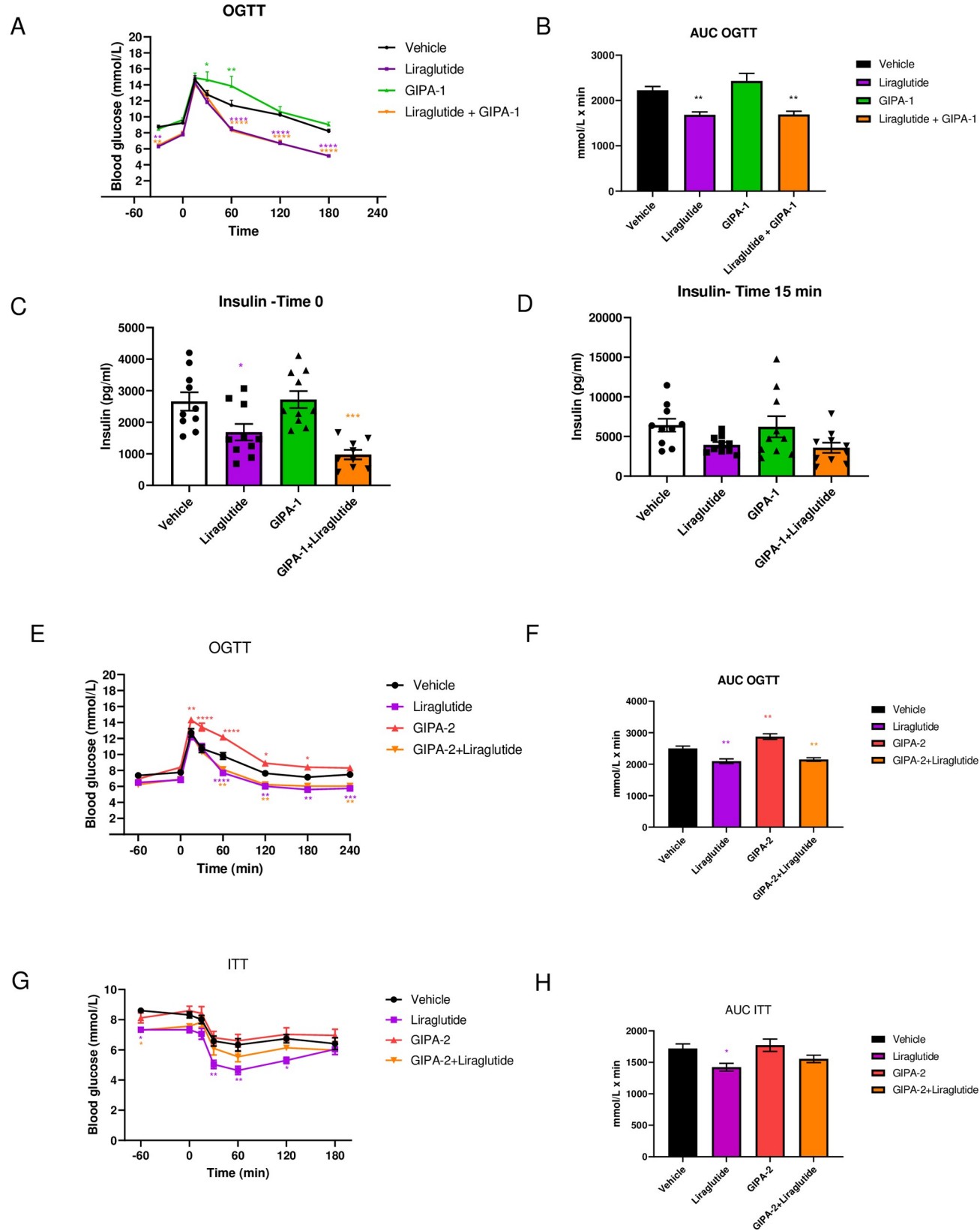

**Fig 3. Effects of GIPA-1 and GIPA-2 on oral glucose tolerance and insulin responses in DIO mice.** Oral glucose tolerance test (A), the corresponding glucose AUC (B) and insulin responses at 0 min (C) and 30 min (D) after the glucose bolus, following administration of vehicle, 0.2 mg/kg liraglutide, ~4.5 mg/kg GIPA-1 and combination of both for 21 days in DIO mice.(E-H) Effects on oral glucose tolerance test (E and F) on day 17, and insulin tolerance test (G and H) on day study 24, following chronic administration of vehicle, 0.2 mg/kg liraglutide, 10 mg/kg GIPA-2 and combination of both in DIO mice. Group sizes are n = 10, and data are represented as mean ± SEM. Statistical analysis was calculated using two-way ANOVA with Tukey's post-hoc test and one-way ANOVA with Dunnett's post-hoc test. *: $p < 0.05$, **: $p < 0.01$, ***: $p < 0.001$, ****: $p < 0.0001$ compared to vehicle.

## Discussion

The identification of novel treatments to combat the burgeoning obesity and diabetes crisis is an urgent health priority. Considerable attention has focused on the physiological actions of gut hormones to control blood glucose, food intake and adiposity. The GLP-1 drug class has emerged as one of the most efficacious and widely-used treatment for diabetes in addition to demonstrated effects for weight loss. However, T2D patients treated with GLP-1 mimetics still experience disease progression eventually requiring insulin therapy, and extensive efforts to improve the use of gut hormone-based therapy are focused on the rational design of combinations that are superior to monotherapy due to synergistic or additive effects on glucose control and weight loss.

The GLP-1-agonist-GIPR-antagonist combination therapy reported in the present study was born from evidence of compromised gut-hormone function in obesity and diabetes as well as their postulated role in driving the beneficial metabolic effects following bariatric surgery. Indeed, GLP-1 secretion is enhanced after bariatric surgery [17] which is thought to be a key component for the resolution of diabetes. In contrast, the controversial role of the GIP signalling pathway in dysmetabolic states has led to the pursuit of both agonist and antagonist approaches [23, 58]. There is a compelling body of evidence suggesting that antagonism of this system can produce beneficial effects on adiposity and insulin sensitivity [38, 40, 45, 46]. The combination of the reported attributes afforded by GIPR antagonism as well as GLP-1's well-established metabolic benefits offered a rationale for investigating this dual therapy in the current study. To explore the potential of this combination, we selected two distinct and pharmacologically-characterised GIPR antagonists for evaluation, GIPA-1 (mouse GIP(3–30)NH$_2$) and GIPA-2 (N$^\alpha$Ac-K10[γEγE-C16]-Arg18-hGIP(5–42)). Since the GIPR antagonist, (Pro3)-GIP, was originally characterized as an antagonist but was subsequently found to also possess partial agonist properties [59], we felt it prudent to characterise both the antagonist and agonist properties of these compounds. To our knowledge, no previous studies have determined the potency of mouse GIP(3–30)NH$_2$ antagonism at the mouse GIPR. In our hands mouse GIP(3–30)NH$_2$ (GIPA-1) did not display agonist behaviour and inhibited mouse GIP-induced activation of mouse GIPR with an IC$_{50}$ of 483 nM. GIPA-2 was a significantly more potent antagonist at mouse GIPR with an IC$_{50}$ of 3 nM and with no evidence of agonism, confirming the findings of Mroz *et al.* [50].

In agreement with our *in vitro* data, GIPA-2 acutely blocked the endogenous GIP incretin effect, impaired glucose tolerance, and inhibited the improved glucose tolerance elicited by exogenously administered native GIP. In contrast, GIPA-1 had no effect on glucose tolerance. We suspected the apparent lack of effect could be due to the short circulating half-life of GIPA-1; the human peptide sequence has a plasma elimination half-life of 7.7± 1.4 minutes in humans [48]. Given the strong *in vitro* pharmacological data and our interest in other metabolic parameters such as adiposity and insulin sensitivity we proceeded with GIPA-1 in our long term DIO studies but delivered the compound using an osmotic minipump to maintain sufficient plasma concentrations for durable GIPR antagonism given its *in vitro* activity. For GIPA-2 we used daily subcutaneous injection based on its extended plasma half-life and *in vivo* duration of action, as previously described [50].

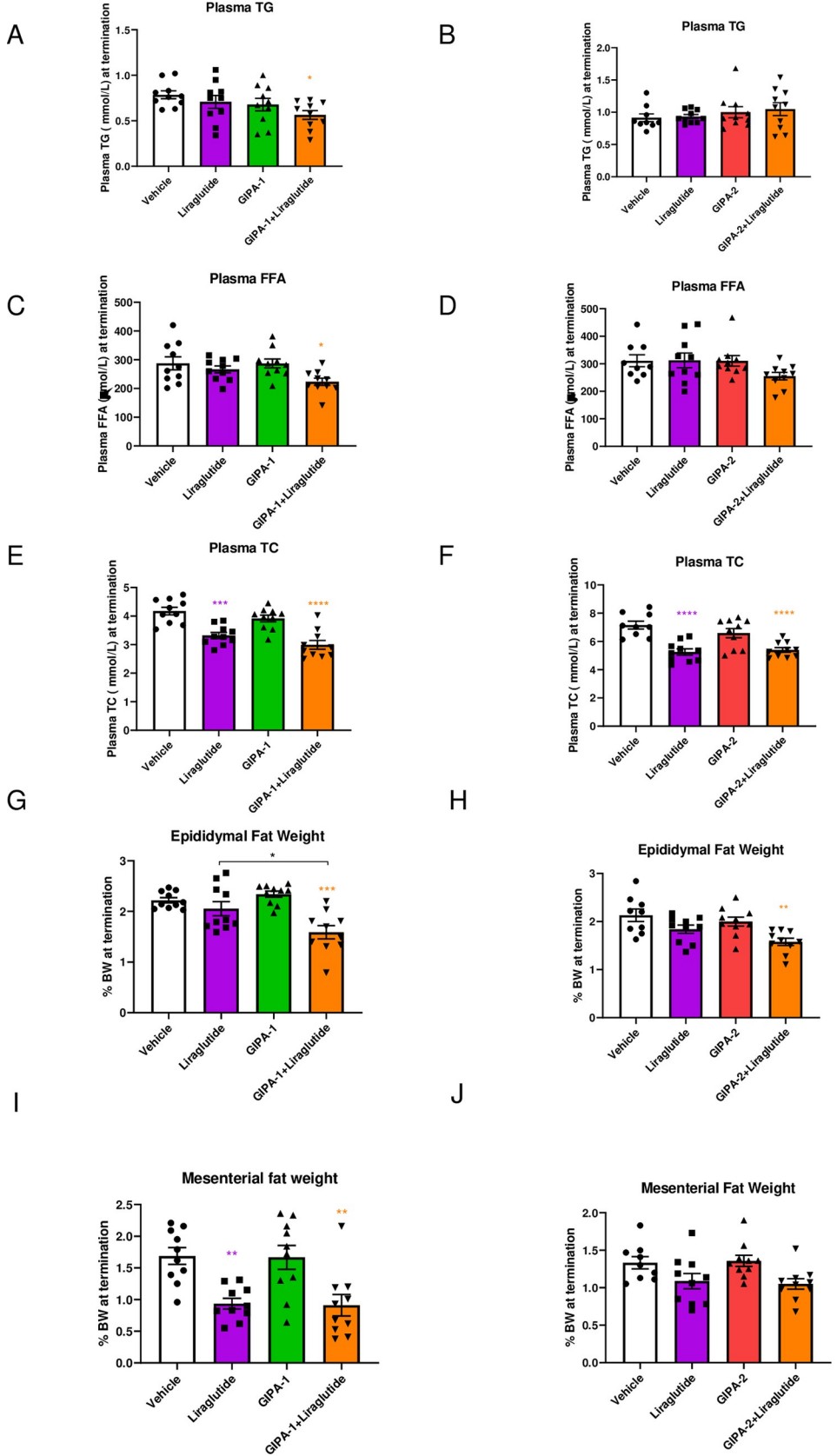

**Fig 4. The effect of GIPR antagonism alone or in combination with liraglutide on lipid homeostasis in DIO mice.**
Effects of GIPA-1 and GIPA-2 on plasma TG (A and B), FFAs (C and D), TC (E and F), epididymal fat weight (G and
H) and mesenteric fat weight (I and J) following administration of vehicle, liraglutide, GIPR antagonist, and liraglutide
in combination with the GIPR antagonist for 28 days in DIO mice. Group sizes are n = 10, and data are represented as
mean ± SEM. Statistical analysis was calculated using one-way ANOVA with Dunnett's post-hoc test. *: p< 0.05, **:
p < 0.01, ***: p < 0.001, ****: p<0.0001 compared to vehicle.

This is the first description of chronic delivery of mouse GIP(3–30)$NH_2$ (GIPA-1) to DIO
mice, alone or in combination with liraglutide. GIPA-1 did not alter body weight or dietary
consumption. However, its combination with liraglutide led to improvements in metabolic
parameters including reductions in epididymal fat weight, plasma FFA, and HOMA-IR, as well
as showing trends in reduced total TG and TC concentrations, when compared with liraglutide
monotherapy. Of note, when administered alone, GIPA-1 negatively impacted glucose control
during a glucose challenge but had no effect when administered together with liraglutide. Taken
together, our data indicate that inhibition of GIPR activity with GIPA-1 in combination with
GLP-1R agonism is associated with modest metabolic benefits including reduced insulin con-
centrations and improved insulin sensitivity without compromising glucose control.

Similarly to GIPA-1, GIPA-2 monotherapy had no effect on body weight or dietary con-
sumption in agreement with previous reports, albeit delivered at a lower dose and shorter time
period than in our study [50]. Co-administration of GIPA-2 with liraglutide significantly
reduced epididymal fat weight and HOMA-IR and tended to reduce plasma FFA, when com-
pared with liraglutide alone. Importantly, despite having a potentiating effect on glucose
excursion during an OGTT when given alone, GIPA-2 did not inhibit liraglutide's beneficial
glucose-lowering effect when administered in combination, mirroring our GIPA-1 data. It
should be noted that due to the nature of GIPA-2 administration (daily injections), OGTT
results might be influenced by both the 'chronic' and 'acute' effects of GIPA-2. The effect of
treatments on different fat pad depots was generally mixed, and combination therapy provided
no extra benefit over liraglutide apart from reducing epididymal fat pad weight.

Taken together our data suggest that combined GLP-1 agonism and peptide based GIP
antagonism produces relatively modest metabolic benefits over GLP-1 given alone, highlighted
by our results showing moderate improvements in fasting glucose, HOMA-IR and the weight
of some fat depots due to combination therapy. However, these benefits are not robust with
the antagonists and methodologies used in the current study. Further optimised peptide antag-
onists may produce stronger effects, more akin to those seen with GIPR antibody approaches.
However, the question of how best to target the GIP system remains enigmatic. Why do both
GIP agonist and antagonist approaches potentially enhance the metabolic improvements seen
with GLP-1 mimetics? Early clinical results for a dual agonist peptide for both the GLP-1 and
GIP receptors support a path forward for GIPR agonist approaches for treating T2D and/or
obesity [23, 53]; however, pre-clinical data from multiple independent studies also validate a
clinical strategy that employs a GIPR antagonist [38, 40, 45, 46]. A recent review by Holst ele-
gantly describes this conundrum and presented several possible scenarios focused around GIP
receptor desensitisation, internalisation and post receptor signalling recruitment leading to
GIP resistance [58]. In a study published after Holst's review, Killion et al. showed that chronic
GIPR agonism desensitizes GIPR activity in primary adipocytes and functions like a GIPR
antagonist, favouring the GIP receptor desensitisation scenario [60]. Importantly, GIP antago-
nists appear able to restore the cell surface expression of GIPR [61] and therefore possess the
potential to restore endogenous GIP sensitivity, a function lost in T2DM. This has direct
implications for our studies. The available GIPR antagonists likely display varied potencies for
GIPR, and responsiveness to endogenous GIP, and the GIPR antibody strategy is presumably

the most potent. However, the challenge remains; how to engineer the balance between pharmacological GIPR antagonism and enhanced endogenous GIP sensitivity to thereby generate a metabolic benefit, and can this balance be more precisely controlled? These are questions which will require further investigation and may result in GIPR antagonists which are more suited for deploying in combination with GLP-1 agonists.

## Supporting information

**S1 Fig.** Representative concentration response curves for stimulation of cAMP accumulation by GIPA-1 (A) and GIPA-2 (B) in CHO-K1 cells expressing mouse GIPR, showing that neither is acting as partial agonist.
(TIF)

**S2 Fig. The effect of GIPR antagonists alone or in combination with liraglutide on lipid homeostasis in DIO mice.** Effects of GIPA-1 and GIPA-2 on liver TG (A and B), liver weight (C and D), retroperitoneal fat weight (E and F) and inguinal fat weight (G and H) following administration of vehicle, liraglutide, GIPR antagonist, and liraglutide in combination with the GIPR antagonist for 28 days in DIO mice. Group sizes are n = 10, and data are represented as mean ± SEM. Statistical analysis was calculated using one-way ANOVA with Dunnett's post-hoc test. *: $p < 0.05$, **: $p < 0.01$, ***: $p < 0.001$, ****: $p < 0.0001$ compared to vehicle.
(TIF)

## Acknowledgments

The authors would like to thank Jason Huang at Fractyl for project management support, Jay Caplan and Emily Cozzi at Fractyl for critical review of this manuscript, and Gitte Hansen, Mette Østergaard, Bidda Rolin, and Cecilia Ratner at Gubra for supporting rodent studies. Authors will adhere to PLOS ONE policies on the sharing of data and materials.

## Author Contributions

**Conceptualization:** Jason A. West, Soumitra S. Ghosh, David G. Parkes, Philip J. Pedersen, David Maggs, Harith Rajagopalan.

**Formal analysis:** Jason A. West, Anastasia Tsakmaki, Soumitra S. Ghosh, David G. Parkes, Philip J. Pedersen, David Maggs, Harith Rajagopalan, Gavin A. Bewick.

**Investigation:** Jason A. West, Anastasia Tsakmaki, Soumitra S. Ghosh, David G. Parkes, Rikke V. Grønlund, Philip J. Pedersen, David Maggs, Harith Rajagopalan, Gavin A. Bewick.

**Methodology:** Jason A. West, Soumitra S. Ghosh, David G. Parkes, Philip J. Pedersen, David Maggs, Harith Rajagopalan.

**Resources:** Jason A. West.

**Writing – original draft:** Jason A. West, Anastasia Tsakmaki, Gavin A. Bewick.

**Writing – review & editing:** Jason A. West, Anastasia Tsakmaki, Soumitra S. Ghosh, David G. Parkes, David Maggs, Harith Rajagopalan, Gavin A. Bewick.

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
