## [Decision Letter · Decision Letter 0]

18 Jan 2021

PONE-D-20-36396

Chronic peptide-based GIP receptor inhibition exhibits modest metabolic changes in mice when administered either alone or combined with GLP-1 agonism

PLOS ONE

Dear Dr. Bewick,

Thank you for submitting your manuscript to PLOS ONE. After careful consideration, we feel that it has merit but does not fully meet PLOS ONE’s publication criteria as it currently stands. Therefore, we invite you to submit a revised version of the manuscript that addresses the points raised during the review process.

You will note that the Reviewers are asking for more clarity with regards to the novelty of the adopted experimental approach, alongside additional information on the experiments employed.

We look forward to receiving your revised manuscript.

Kind regards,

Nigel Irwin

Academic Editor

PLOS ONE

Journal Requirements:

2. As part of your revisions please address the following items pertaining to animal care and welfare considerations:

(i) monitoring parameters during the study (the criteria used to evaluate the health and well-being of the animals; (ii) a proper explanation about the humane endpoints that you had in place for any animal who became severely ill/injured prior to the experimental endpoint;

(iii) rate of mortality during the study, and details about humane endpoints for any animals who became severely ill prior to the experimental endpoint;

(iv) a description of anesthetics and analgesics administered to animals during your study (name of drug, dosage, frequency and route of administration) - if you have already included this info, please disregard this request;

(v) method of euthanasia.

3. Thank you for including your ethics statement:  "All animal experiments were performed according to the bioethical guidelines of Gubra (Hørsholm, Denmark) under the personal license 2017-15-0201-01378, which was issued by the Danish Animal Experimentation Council. ".   

a. Please amend your current ethics statement to confirm that your named ethics committee specifically approved this study.

For additional information about PLOS ONE submissions requirements for ethics oversight of animal work, please refer to http://journals.plos.org/plosone/s/submission-guidelines#loc-animal-research  

4. Thank you for providing the following Funding Statement: 

'This study was supported by Fractyl Laboratories Inc., Lexington, MA'

a. We note that one or more of the authors is affiliated with the funding organization, indicating the funder may have had some role in the design, data collection, analysis or preparation of your manuscript for publication; in other words, the funder played an indirect role through the participation of the co-authors.

If the funding organization did not play a role in the study design, data collection and analysis, decision to publish, or preparation of the manuscript and only provided financial support in the form of authors' salaries and/or research materials, please review your statements relating to the author contributions, and ensure you have specifically and accurately indicated the role(s) that these authors had in your study in the Author Contributions section of the online submission form. Please make any necessary amendments directly within this section of the online submission form. 

Please also update your Funding Statement to include the following statement: “The funder provided support in the form of salaries for authors [insert relevant initials], but did not have any additional role in the study design, data collection and analysis, decision to publish, or preparation of the manuscript. The specific roles of these authors are articulated in the ‘author contributions’ section.”

If the funding organization did have an additional role, please state and explain that role within your Funding Statement.

We note that one or more of the authors are employed by commercial companies: Doon Associates LLC, DGP Scientific Inc, Gubra ApS and Becton Dickinson Technologies & Innovation.

Please also declare these commercial affiliations in your amended Funding Statement, as well as a statement regarding the Role of Funders in your study. If the funding organization did not play a role in the study design, data collection and analysis, decision to publish, or preparation of the manuscript and only provided financial support in the form of authors' salaries and/or research materials, please review your statements relating to the author contributions, and ensure you have specifically and accurately indicated the role(s) that these authors had in your study. You can update author roles in the Author Contributions section of the online submission form.

b. Please also provide an updated Competing Interests Statement declaring these commercial affiliations along with any other relevant declarations relating to employment, consultancy, patents, products in development, or marketed products, etc.  

Reviewers' comments:

Reviewer's Responses to Questions

**Comments to the Author**

1. Is the manuscript technically sound, and do the data support the conclusions?

Reviewer #1: Yes

Reviewer #2: No

2. Has the statistical analysis been performed appropriately and rigorously? 

Reviewer #1: Yes

Reviewer #2: Yes

3. Have the authors made all data underlying the findings in their manuscript fully available?

Reviewer #1: Yes

Reviewer #2: Yes

4. Is the manuscript presented in an intelligible fashion and written in standard English?

Reviewer #1: Yes

Reviewer #2: Yes

5. Review Comments to the Author

Reviewer #1: This work by West and colleagues investigates the effect of two independent GIPR antagonist, along and in combination with liraglutide, on metabolic outcomes. The follow a similar experimental plan that has been done previous with unique GIPR antagonists (Svendsen et al, Killion et al) and with the GIPA-2 (Mroz et al). The overall conclusion is that GIPR antagonism alone provides limited to no effect on body weight or other metabolic outcomes, which largely agrees with previous studies. The combination with liraglutide in the current studies is not additive, which agrees with the studies by Svendsen and Mroz. The antagonists are not new, GIP(3-30) has been extensively characterized by the Danish group and the GIPA-2 has been published by Mroz. Overall, there is very little no information here, as the studies are mostly confirmatory. Moreover, the ongoing debate of the usefulness of GIP(3-30) is not address here, as this antagonist seems to have very little effect in vivo. It remains unclear why the studies of Killion show additivity with GLP-1R agonism (there is a second manuscript recently published in Nature Communications that is not referenced), while other GIPR antagonists do not. Moreover, whether this strategy holds promise beyond mice is unclear.

- Introduction – 4th paragraph. Evidence is provided documenting how loss of function studies for the GIPR protect against DIO in preclinical models. This is extended to state that “GIP plays a pathophysiological role in the development of obesity and insulin resistance”. There is no evidence that increasing GIPR activity drives obesity or insulin resistance – in fact, all pharmacology and genetic models produce modest decreases in body weight. This is a common misconception in this field and should be correct here.

- Introduction – 5th paragraph. Gipg013 is written as Gipg103 twice.

- Methods – GIPA-2 is supported by reference 58. This is a review article on GIP, but does not contain explicit (or any?) information on this peptide. It seems this peptide antagonist is described in the work by Mroz, Mol Metabolism (ref 50)?

- The units in Figure 1A and B are missing in the figure and legend

- With an IC50 of 483 nM and no evidence of antagonist properties in vivo, how can the authors be confident that they are achieve sufficient antagonism of the GIPR with GIPA-1? This is particularly important since the majority of work documenting GIP(3-30) has come from a single group and most of this evidence is in vitro with non-physiological models. Confirmation that this reagent is a truly a useful GIPR antagonist is required. Compelling evidence is provided for GIPA-2 (Figure1 E-J). Would not the conclusion from Figure 1 be that GIPA-1 is not an effective antagonist in vivo?

Reviewer #2: West et al. examined the effect of GIPR peptide antagonists on energy- and glucose-metabolism and found that the GIPR peptide antagonists used in this study exhibited negligible effect on body weight and glucose metabolism. Since the therapeutic efficacy of the GIP antagonist alone or with combination with GLP-1RAs is an important topic to be clarified, the current study is of great impact to the readers, regardless of their conclusion (i.e., whether they are effective or not). However, this paper seems to include half-baked conclusions; ‘lack of effect on metabolic changes’ and ‘beneficial effect on plasma TG and FFA when co-administered with GLP-1 agonist’. Unfortunately, the authors failed to demonstrate either of the concepts, because of the flaws in their experimental designs.

If the authors intend to focus on the lack of effect of GIPR peptide antagonists, they had to show that their chronic treatment (the dose and the way of administration) with GIPA-1 or GAPA-2 is sufficient in blocking the GIP signaling in vivo. By contrast, if the authors intend to propose that GIPR peptide antagonists are effective in lowering HOMA-IR, plasma TG, plasma FFA, and the weight of epididymal fat depots when combined with liraglutide, they had to show that any alteration in triglyceride metabolism in epididymal fat depots or in plasma was induced by the treatment. Unfortunately, such evaluation is not included at all in the present paper and the lack of data makes the paper less attractive.

In addition, the current paper includes many inappropriate protocols, which hamper the precise evaluation of the efficacy of GIPA-1 or GAPA-2 during OGTT and ITT as shown below.

<major comments="">

1. Inappropriate protocol for OGTT and ITT in the mice treated chronically with GIPA-1 or GAPA-2 (Fig. 3)

In the Fig. 3, the authors treated the mice with GIPA-1, GAPA-2, and/or liraglutide at 30 or 60 min before OGTT and ITT. As for Figs. 3A-D, the mice had been treated for 4 weeks with GIPA-1 (using the osmotic pomp) and/or liraglutide (by daily s.c. injection). Therefore, no further pre-administration of GIPA-1 is required for evaluating its chronic effect on the day of OGTT (on Day 21). In addition, as for Figs. 3E-H, the mice had been treated for 4 weeks with GIPA-2 and/or liraglutide (both by daily s.c. injection). On the day of OGTT and ITT (on Day 21), the daily dose of GIPA-2 and/or liraglutide was administered at 60 min before the tests. Apparently, this protocol is inappropriate for evaluating the chronic effect of GAPA-2. To circumvent this problem, GIPA-2 should have been also given continuously using an osmotic pump.

2. Lack of data on the mechanism of the alteration in HOMA-IR, plasma TG, plasma FFA, and epididymal fat weight by GIPA-1 (Fig.2J and Fig.4)

Alteration in the levels of plasma TG, plasma FFA, and epididymal fat weight induced by chronic treatment with GIPA-1 is not so drastic, but these results may imply important insight on the role of GIP on fat metabolism. Reduction in HOMA-IR by GIPA-1 (Fig. 3J) may also be potentially interesting. However, their mechanisms have not been examined in the present study. GIP has been suggested to participate in promoting TG accumulation in fat tissues. Therefore, it is of great importance to show whether the chronic treatment with GIPR peptide antagonists impacts these parameters. LPL activity in the plasma, HSL phosphorylation in the fat tissues, gene expressions of lipogenic genes and lipolytic genes in several fat depots, and inulin signaling in epididymal fat tissues would be required for evaluating their underlying mechanisms.

Difference in tissue weight change between different fat depots is also interesting. The expression levels of GIPR in different fat depots after chronic treatment may be of interest. In addition, the changes in GIPR expression in bone by the treatments are also interesting to be checked, considering that the GIP antagonism may have deteriorating effect on bone homeostasis.

<minor comments="">

1. Ambiguous conclusion in the title

In the title, they say GIP antagonists showed ‘modest’ effect on metabolism. However, in the abstract, they focused on their positive effect on insulin sensitivity and plasma TG and FFA levels. The message of this paper should be clarified. The third paragraph in the abstract is redundant with the latter part of the second paragraph.

2. 4 hour fasting before OGTT (Figs.3A, E)

The mice seemed to be fasted only for 4 hours before OGTT. The reviewer feels this somewhat strange. Is there any special reason why they were not fasted overnight (~16hr) before OGTT. Some statements would better be added.</minor></major>

6. PLOS authors have the option to publish the peer review history of their article (what does this mean?). If published, this will include your full peer review and any attached files.

Reviewer #1: No

Reviewer #2: **Yes: **Takashi Miki

---

## [Author Response · Author response to Decision Letter 0]

24 Feb 2021

Journal Requirements:

We believe we have followed all of PLOS ONE's style requirements.

2. As part of your revisions please address the following items pertaining to animal care and welfare considerations:

(i) monitoring parameters during the study (the criteria used to evaluate the health and well-being of the animals; 

When a new study starts up, a kick-off meeting is held where all personnel involved in the study attend. This is to align activities on the specific study and to create expectations regarding the animals and highlight extra attention to any potential complications in the study period. 

The arrangement for providing care in the studies was set up in a manner where all animals were checked at least daily, but clinical surveillance was more frequent (up to three times during the day) on the day when compound dosing was initiated (especially for animals with an implanted osmotic minipump also recovering from surgery). 

The animal monitoring consists of daily visual inspections by the Animal Caretakers, regular or daily weight monitoring, and for some animals food intake measurements during chronic dosing studies. 

The animal care programme ensures that all animals that require special or medical attention are provided with care as soon as possible. If an animal caretaker or a scientist observe any animals showing signs of illness or distress, they are instructed to immediately contact the Attending Veterinarian for a health evaluation. Animal caretakers are instructed to look for a variety of pathological signs and abnormalities during their regular handling of the animals. These include activity, hydration, body condition score, teeth, ulcerations on the ears and in the neck area, mucous plugs, etc. Incorporated in the weight system (HM View) is a warning that gives the Animal Caretakers who weigh the animals a notice if the animal has lost a certain amount of weight over the last few days (mice ±10% or 3 g. deviation between last and current body weight). This ensures that animals without clinical or external signs of disease will be assessed based on differences in weight. Furthermore, weight and food intake data are uploaded directly to our inhouse data explorer “GubraView” which gives both study directors and veterinarians an excellent tool to monitor the animals, as small deviations in weight and food intake will stand out and make it possible for the veterinary staff to take necessary action.

If animals are found in a critical condition in the study period (serious or severe side effect), both animal caretakers and scientists are instructed to euthanize the affected animal without further delay (se humane endpoints below). 

We have added additional language to the manuscript to emphasize the key points covered above. 

(ii) a proper explanation about the humane endpoints that you had in place for any animal who became severely ill/injured prior to the experimental endpoint;

The specific assessment criteria for humane endpoints used are always based on a veterinary judgement of the animal’s well-being, and before initiation of studies, a meeting between a veterinarian and the study director must be held to discuss the animal model being used in the study and to evaluate potential risks. Following, the Study Director hosts a kick-off meeting with the Principal Technician (Animal Caretaker) and other people involved in the study conduction. Here the humane endpoints are passed on to the Animal Caretakers.

All studies conducted at Gubra adhere to a specific license approving the experiment and animal model. Herein the study-specific humane endpoints are listed and approved by the authorities (the Danish Animal Experimentation Council). 

The following humane endpoints were used for the current studies: 

• Rapid weight loss of ≥20 percent within a few days (<5).

• Gradual weight loss over a longer period leading to emaciation (the limit is 20 percent below the weight of a normal healthy control animal of the same species and age). 

• Clinical or behavioral signs such as inactivity and loss of interest in the surroundings.

• Inability to access food or water.

• Forced abdominal respiration.

• Dehydration leading to reduced skin elasticity, skin pinched over the back should return to its normal position after it is released.

• The presence of deep open wounds or large tumors.

• Swelling, redness and/or pain response from tissue on and around the osmotic pump.

• Dehiscence (splitting apart) of surgical incisions from pump implantations which cannot be re-sutured or discharge from surgical incisions.

• Any condition or test compound indicated to cause suffering in the animals.

• Local reactions to compound injections causing inflammation or wounds of the cutis.

Any other adverse reactions to the compound resulting in any of the above-mentioned signs of pain and distress.

We have also added additional language to the manuscript to cover the key points outlined above. 

(iii) rate of mortality during the study, and details about humane endpoints for any animals who became severely ill prior to the experimental endpoint;

One animal from the vehicle group was found dead in the GIPA-2 study on study day 28. At necropsy the animal was found have a liver tumor, which as attributed to both the age of the animal (52 weeks) and the DIO diet. 

We have added this detail to the manuscript. 

(iv) a description of anesthetics and analgesics administered to animals during your study (name of drug, dosage, frequency and route of administration) - if you have already included this info, please disregard this request;

Osmotic minipump used in the GIPA-1 study were implanted under isoflurane/O2 inhalation anaesthesia (isoflurane 2-3%). Animals were treated with analgesics on day 0-3 (NSAID, carprofen 50 mg/kg SC QD). 

We added these details to the manuscript. 

(v) method of euthanasia.

Animals included in the Acute in vivo studies were euthanized by cervical dislocation. Animals included in the GIPA-1 and GIPA-2 studies were euthanized be cardiac bleeding under isoflurane/O2 inhalation anaesthesia. 

Again, this is now included in the manuscript. 

 3. Thank you for including your ethics statement: "All animal experiments were performed according to the bioethical guidelines of Gubra (Hørsholm, Denmark) under the personal license 2017-15-0201-01378, which was issued by the Danish Animal Experimentation Council. ". 

a. Please amend your current ethics statement to confirm that your named ethics committee specifically approved this study.

All animal experiments were performed according to the bioethical guidelines of Gubra (Hørsholm, Denmark) under the personal license 2017-15-0201-01378, which was issued by the Danish Animal Experimentation Council. The initial study designs submitted for licensing and the post-approval monitoring of the animal care and the regulatory compliance was the responsibility of the Gubra IACUC.

For additional information about PLOS ONE submissions requirements for ethics oversight of animal work, please refer to http://journals.plos.org/plosone/s/submission-guidelines#loc-animal-research

Ethic statement has been amended.

4. Thank you for providing the following Funding Statement: 'This study was supported by Fractyl Laboratories Inc., Lexington, MA'

a. We note that one or more of the authors is affiliated with the funding organization, indicating the funder may have had some role in the design, data collection, analysis or preparation of your manuscript for publication; in other words, the funder played an indirect role through the participation of the co-authors.

Employees and shareholders of Fractyl Laboratories Inc played a prominent role in the study design, data collection and analysis, decision to publish, and preparation of the manuscript. Conceptualization and methodology: J.A.W., S.S.G., D.G.P., P.J.P., D.M., and H.R.; Investigation: J.A.W., A.T., S.S.G., D.G.P, R.V.G., P.J.P, D.M., H.R, G.A.B.; Formal analysis and writing, review, and editing: J.A.W., A.T., S.S.G., D.G.P., P.J.P, D.M., H.R, G.A.B. All authors reviewed and critiqued the manuscript throughout the editorial process, approved the final manuscript draft submitted for publication, and agreed to be accountable for all aspects of the work, ensuring the accuracy and integrity of the publication.

b. Please also provide an updated Competing Interests Statement declaring these commercial affiliations along with any other relevant declarations relating to employment, consultancy, patents, products in development, or marketed products, etc. 

J.A.W. and H.R. and are employees and shareholders of Fractyl Laboratories Inc. A.T. has received funding/grant support from the Juvenile Diabetes Research Foundation (JDRF). S.S.G. is an employee of Doon Associates and has received honorariums for consultancy from Fractyl Laboratories Inc. D.G.P. is an employee of DPB Scientific and has received honorariums for consultancy from Fractyl Laboratories Inc R.V.G. and P.J.P. are employees of Gubra ApS. D.M. is an ex-employee of Fractyl Laboratories Inc., is a current shareholder, and has received honorarium for consultancy from Fractyl Laboratories Inc. G.A.B. has received funding/grant support from the European Foundation for the Study of Diabetes and JDRF and honorarium for consultancy from Fractyl Laboratories Inc. This does not alter our adherence to PLOS ONE policies on the sharing of data and materials.

c. Please include both an updated Funding Statement and Competing Interests Statement in your cover letter.

Please see updated cover letter

Reviewers' comments:

Reviewer 1 

Thank you very much for taking the time to review our manuscript. We welcome your comments and have addressed the points you raised below and where appropriate updated the manuscript.

As you correctly stated the antagonists, have been reported previously. GIPA-2 was used by Mroz et al. However, we not only provide independent confirmatory data for this antagonist but importantly extend it with novel data we believe is of value to the field. In a longer chronic study we add to the body weight data reported by Mroz et al. to include food intake, fasting blood glucose and insulin, HOMA-IR, OGTT, ITT, plasma TG, TC and FFA and weight of various fat depots. 

In regard to GIPA-1, to our knowledge, no previous studies have determined the potency of mouse GIP(3-30)NH2 antagonism at the mouse GIPR. Moreover, it has not been published in combination with liraglutide. Even though chronic delivery of GIPA-1 alone had negligible effects on food intake, body weight, fasting blood glucose and plasma insulin concentrations, when delivered in combination with liraglutide it augmented liraglutides insulin sensitizing effects and lowered plasma triglycerides and free-fatty acids. We believe both these sets of data are of use to the field and suggest GIPA-1 may not be useful alone at least in the context we delivered it, but it could have potential in accruing benefit in combination with other compounds. However, it is clear there is further work to be done to adequately resolve the remaining controversies and identify mechanisms of action. We also agree that the available data in the literature does not currently provide a convincing pathway to translation, as yet. 

- there is a second manuscript recently published in Nature Communications that is not referenced. 

We apologise for this oversight. It has now been included in our references and been part of our discussion. The text now reads: “In a study published after Holst’s review, Killion et al. showed that chronic GIPR agonism desensitizes GIPR activity in primary adipocytes and functions like a GIPR antagonist, favouring the GIP receptor desensitisation scenario [60]”.

- Introduction – 4th paragraph. Evidence is provided documenting how loss of function studies for the GIPR protect against DIO in preclinical models. This is extended to state that “GIP plays a pathophysiological role in the development of obesity and insulin resistance”. There is no evidence that increasing GIPR activity drives obesity or insulin resistance – in fact, all pharmacology and genetic models produce modest decreases in body weight. This is a common misconception in this field and should be correct here.

The reviewer is correct, we have probably overinterpreted the current available data. We have deleted this statement; the text now reads:

“All the above evidence warrants further investigation and has led to discovery programs focused on blocking GIP activity”.

- Introduction – 5th paragraph. Gipg013 is written as Gipg103 twice.

Thank you for pointing out our mistake. It has now been corrected throughout the manuscript.

- Methods – GIPA-2 is supported by reference 58. This is a review article on GIP, but does not contain explicit (or any?) information on this peptide. It seems this peptide antagonist is described in the work by Mroz, Mol Metabolism (ref 50)?

This mistake has now been corrected.

- The units in Figure 1A and B are missing in the figure and legend

 Concentration of each peptide was measured in uM. It is now clearly stated in the figure legend.

- With an IC50 of 483 nM and no evidence of antagonist properties in vivo, how can the authors be confident that they are achieve sufficient antagonism of the GIPR with GIPA-1? This is particularly important since the majority of work documenting GIP(3-30) has come from a single group and most of this evidence is in vitro with non-physiological models. Confirmation that this reagent is a truly a useful GIPR antagonist is required. Compelling evidence is provided for GIPA-2 (Figure1 E-J). Would not the conclusion from Figure 1 be that GIPA-1 is not an effective antagonist in vivo?

It has been shown that rat and the human GIP(3-30)NH2 are effective antagonists in vivo [48, 49], so we would expect the mouse GIP(3-30)NH2 to behave in a similar manner. In rats, subchronic treatment with GIPR antagonist, rat GIP (3-30)NH2, did not modify food intake or bone resorption, but significantly increased body weight, body fat mass, triglycerides, LPL, and leptin levels compared with vehicle treated rats (ref. 49)(dosing: 25 nmol/kg b.w. for 3 weeks)[49]. In humans GIP-induced insulin secretion was reduced by 82% during co-infusion with GIP(3-30)NH2 [48].

No previous studies have determined the potency of mouse GIP(3-30)NH2 antagonism at the mouse GIPR. We agree that an IC50 of 483 nM is a bit high indicating that the mouse GIP(3-30)NH2 is probably moderately active and in our hands it had no effect on glucose tolerance in an acute in vivo study. However, as we mentioned in the discussion, we suspected the apparent lack of effect could be due to the short circulating half-life of GIP(3-30)NH2; the human peptide sequence has a plasma elimination half-life of 7.7± 1.4 minutes in humans [48]. Given the strong in vitro pharmacological data and our interest in other metabolic parameters such as adiposity and insulin sensitivity we proceeded with GIP(3-30)NH2 in our long term DIO studies but delivered the compound using an osmotic minipump to maintain sufficient plasma concentrations for durable GIPR antagonism given its in vitro activity.

Even though chronic delivery of mouse GIP(3-30)NH2 (GIPA-1) alone had negligible effects on lowering food intake, body weight, fasting blood glucose and plasma insulin concentrations, when used in combination with liraglutide it augmented insulin sensitizing effects and lowered plasma triglycerides and free-fatty acids, showing that it was an active antagonist in vivo with even more notable metabolic effects than GIPA-2, which had a lower IC50. Obviously more work is needed to fine tune the dosing regimen.

Reviewer 2

Thank you very much for taking the time to review our manuscript. We have addressed your comments below.

1. Inappropriate protocol for OGTT and ITT in the mice treated chronically with GIPA-1 or GAPA-2 (Fig. 3)

Apparently, this protocol is inappropriate for evaluating the chronic effect of GAPA-2. To circumvent this problem, GIPA-2 should have been also given continuously using an osmotic pump.

In the GIPA-1 study, GIPA-1 was given continuously using an osmotic pump. No GIPA-1 was administered before the OGTT. Please forgive our typographical mistakes. Liraglutide and vehicle were administered by subcutaneous injection 30 minutes prior to the OGTT on day 21. We have now corrected the manuscript, it now reads:

Material and Methods (GIPA-1 study): On the day of the OGTT the fasted animals received their daily dose of vehicle, and liraglutide and GIPA-1 thirty minutes before receiving a glucose bolus.

In the GIPA-2 study an OGTT was performed on day 17 and an ITT on day 24. On both occasions fasted animals received their daily dose of vehicle, liraglutide and/or GIPA-2 one hour before receiving a glucose bolus or insulin, respectively. In this study the peptide antagonist was given by subcutaneous injection once daily at the same time (10 am). The administration of GIPA-2 one hour before the OGTT and ITT on days 17 and 24 maintained this regimen. Clearly it was important not to disrupt chronic daily dosing in the middle of the study. This is standard protocol. Indeed, we also believe that this does not hamper the interpretation of the OGTT results. This paradigm evaluates if GIPA2 continues to be effective at regulating acute glucose tolerance in the context of chronic delivery. At the same time, we evaluated fasting glucose and insulin and calculated longitudinal HOMA-IR as markers of the long-term effects of GIPA-2 on metabolic status. 

We did not consider it necessary to deliver GIPA-2 by osmotic pump as we had shown it to be a very effective inhibitor in our acute in vivo experiment. Moreover, previous studies have delivered this compound in this way [50]. Furthermore, GIPA-2 is an N-terminally truncated form of human GIP that has been site specifically fatty acylated to enable daily subcutaneous dosing and substituted with Arg18 to enhance potency at mGIPR [50]. 

2. Lack of data on the mechanism of the alteration in HOMA-IR, plasma TG, plasma FFA, and epididymal fat weight by GIPA-1 (Fig.2J and Fig.4)

We agree that the effect of the antagonists in combination with liraglutide in lowering HOMA-IR, plasma TG, plasma FFA, and the weight of epididymal fat depots is interesting and requires further investigation to unravel the mechanisms. The role of GIP antagonism in bone homeostasis is also another aspect requiring further investigation. However, addressing these questions properly is beyond the scope of the current manuscript. Our aim was to characterise the gross physiological changes produced by the combination therapy, which we believe was achieved. To investigate the mechanisms driving the observations of interest accurately and deeply would require a significant effort more suited to a future manuscript. 

3. Ambiguous conclusion in the title

In the title, they say GIP antagonists showed ‘modest’ effect on metabolism. However, in the abstract, they focused on their positive effect on insulin sensitivity and plasma TG and FFA levels.

We do not agree with the reviewer. It was our intention to present a balanced view of our results which offer some negative data mixed with small beneficial effects on some metabolic parameters. By doing this we have avoided unnecessary hyperbole and presented an honest account. The message is clear; in our hands GIP antagonists do not produce large effects on metabolism, they can be described as modest at best. We highlight the subtle positive effects in the abstract as these are of interest to the reader. 

4.The third paragraph in the abstract is redundant with the latter part of the second paragraph.

Thank you, we have refined the language of our abstract to avoid redundancy. 

5. 4 hour fasting before OGTT (Figs.3A, E)

The mice seemed to be fasted only for 4 hours before OGTT. The reviewer feels this somewhat strange. Is there any special reason why they were not fasted overnight (~16hr) before OGTT. Some statements would better be added.

Historically, the routinely used fasting paradigm for mice was a 14-18h overnight fast with the GTT carried out in the morning after lights-on (8-10am). This establishes a very reproducible plasma glucose baseline but at the expense of most of the liver glycogen stores and is perhaps more representative of starvation than physiological glucose regulation. To account for this, 4-6h hour fasting is now believed to be a much better paradigm. Indeed, a 6h fast provides a more physiological context, preserving liver glycogen and preventing the increase in insulin stimulated glucose utilisation following prolonged fasting in mice. A 6h fast generated clearer data with more statistical significance, when comparing high fat and chow fed C57Blk/6 mice in the following study: Andrikopoulos, S., et al., Evaluating the glucose tolerance test in mice. Am J Physiol Endocrinol Metab, 2008. 295(6): p. E1323-32.

For GTTs performed in studies referenced in our manuscript mice were also fasted for 4-6 h before the performance of GTT. In Kilion et al. for oral glucose tolerance test mice were fasted for 4-6h [40] and in Mroz et.al for 6h [50].

---

## [Decision Letter · Decision Letter 1]

8 Mar 2021

PONE-D-20-36396R1

Chronic peptide-based GIP receptor inhibition exhibits modest metabolic changes in mice when administered either alone or combined with GLP-1 agonism

PLOS ONE

Dear Dr. Bewick,

Thank you for submitting your manuscript to PLOS ONE. After careful consideration, we feel that it has merit but does not fully meet PLOS ONE’s publication criteria as it currently stands. Therefore, we invite you to submit a revised version of the manuscript that addresses the points raised during the review process. You will note that Reviewer 1 is now satisfied with the revised manuscript, with Reviewer 2 suggesting some minor revisions prior to publication. Please simply address the minor comments raised by Reviewer 2.

We look forward to receiving your revised manuscript.

Kind regards,

Nigel Irwin

Academic Editor

PLOS ONE

Journal Requirements:

Reviewers' comments:

Reviewer's Responses to Questions

**Comments to the Author**

1. If the authors have adequately addressed your comments raised in a previous round of review and you feel that this manuscript is now acceptable for publication, you may indicate that here to bypass the “Comments to the Author” section, enter your conflict of interest statement in the “Confidential to Editor” section, and submit your "Accept" recommendation.

Reviewer #1: All comments have been addressed

Reviewer #2: All comments have been addressed

2. Is the manuscript technically sound, and do the data support the conclusions?

Reviewer #1: Yes

Reviewer #2: Yes

3. Has the statistical analysis been performed appropriately and rigorously? 

Reviewer #1: Yes

Reviewer #2: Yes

4. Have the authors made all data underlying the findings in their manuscript fully available?

Reviewer #1: Yes

Reviewer #2: Yes

5. Is the manuscript presented in an intelligible fashion and written in standard English?

Reviewer #1: Yes

Reviewer #2: Yes

6. Review Comments to the Author

Reviewer #1: (No Response)

Reviewer #2: 1. Inappropriate protocol for OGTT and ITT in the mice treated chronically with GIPA-1 or GAPA-2 (Fig. 3)

In regards to the GIGA-1 pre-administration on the day of OGTT, I now understand their protocol. I also confirmed the appropriate correction in the method section in the revision (lines 8-10 on page 9).

I still think the GIGA-2 pre-administration on the day of OGTT and ITT makes the interpretation ambiguous. If the single injection of GIGA-2 per day is sufficient for inhibiting GIP signaling up to the next 24 hours, the authors might have not needed to treat another shot on the day of OGTT or ITT. I now accept their protocol on GIGA-2, but I suggest that the authors may add some statements mentioning that the results might be influenced by both ‘chronic’ and ‘acute (the last shot)’ effect of GIGA-2 administration under this protocol.

2. Lack of data on the mechanism of the alteration in HOMA-IR, plasma TG, plasma FFA, and epididymal fat weight by GIPA-1 (Fig.2J and Fig.4)

I understand the situation, but I think it more important to clarify these phenotypes.

3. Ambiguous conclusion in the title

I think the effect of GIP receptor inhibition on insulin sensitivity, circulating lipids and certain adipose stores may draw much attention through an accumulation of new findings in the future. Lipid metabolism comprises a large part of ‘metabolism’. The current study only showed that GIP receptor inhibition exhibits modest effect on ‘glucose’ metabolism. To make the title more consistent with the abstract, I suggest the authors to add ‘glucose’ in the title.

7. PLOS authors have the option to publish the peer review history of their article (what does this mean?). If published, this will include your full peer review and any attached files.

Reviewer #1: No

Reviewer #2: No

---

## [Author Response · Author response to Decision Letter 1]

12 Mar 2021

Journal Requirements:

Our reference list has been reviewed and we can confirm that no cited paper has been retracted. Only for reference 28, there was a publisher correction involving addition of one of the author’s name omitted from the authors list in the original published article. 

Reviewers' comments:

Reviewer 2

Thank you very much for taking the time to review our manuscript. We welcome your comments and have addressed the points you raised below and where appropriate updated the manuscript.

1. Inappropriate protocol for OGTT and ITT in the mice treated chronically with GIPA-1 or GAPA-2 (Fig. 3)

In your comments for GIPA-2 administration you stated, “If the single injection of GIGA-2 per day is sufficient for inhibiting GIP signaling up to the next 24 hours, the authors might have not needed to treat another shot on the day of OGTT or ITT”. We would like to clarify that on the days of OGTT and ITT animals received only one shot of GIPA-2 (their daily dose, always administrated at the same time as part of their chronic daily dosing).

We understand your concern and we agree that our OGTT and ITT results might influenced by both ‘chronic’ and ‘acute’ effects of GIPA-2 administration under our protocol. To this extend, we have now added a relevant statement in the discussion (5th paragraph, in bold and blue). The text now reads:

“Importantly, despite having a potentiating effect on glucose excursion during an OGTT when given alone, GIPA-2 did not inhibit liraglutide’s beneficial glucose-lowering effect when administered in combination, mirroring our GIPA-1 data. It should be noted that due to the nature of GIPA-2 administration (daily injections), OGTT results might be influenced by both the ‘chronic’ and ‘acute’ effects of GIPA-2.”

2. Lack of data on the mechanism of the alteration in HOMA-IR, plasma TG, plasma FFA, and epididymal fat weight by GIPA-1 (Fig.2J and Fig.4)

As we stated previously, we agree it is important to investigate the molecular mechanisms behind the effects of antagonists in combination with liraglutide in plasma TG, plasma FFA, and the weight of epididymal fat depots to gain a better insight on the role of GIP on fat metabolism. However, we stand behind our previous statement that clarifying these phenotypes properly would require a significant effort and is beyond the scope of the current manuscript. This is now (after your suggestion) also reflected in the manuscript title, stating GIP receptor inhibition exhibits modest effect on glucose metabolism, making it the focus of our study.

3. Ambiguous conclusion in the title

Thank you for your suggestion. We have now added “glucose” in the title. The title now reads: “Chronic peptide-based GIP receptor inhibition exhibits modest glucose metabolic changes in mice when administered either alone or combined with GLP-1 agonism”.

---

## [Editor Report · Decision Letter 2]

15 Mar 2021

Chronic peptide-based GIP receptor inhibition exhibits modest glucose metabolic changes in mice when administered either alone or combined with GLP-1 agonism

PONE-D-20-36396R2

Dear Dr. Bewick,

We’re pleased to inform you that your manuscript has been judged scientifically suitable for publication and will be formally accepted for publication once it meets all outstanding technical requirements.

Kind regards,

Nigel Irwin

Academic Editor

PLOS ONE
---

## [Editor Report · Acceptance letter]

18 Mar 2021

PONE-D-20-36396R2 

Chronic peptide-based GIP receptor inhibition exhibits modest glucose metabolic changes in mice when administered either alone or combined with GLP-1 agonism 

Dear Dr. Bewick:

I'm pleased to inform you that your manuscript has been deemed suitable for publication in PLOS ONE. Congratulations! Your manuscript is now with our production department. 

Kind regards, 

on behalf of

Prof Nigel Irwin 

Academic Editor

PLOS ONE